# Study of Electric and Magnetic Properties of Iron-Modified MFI Zeolite Prepared by a Mechanochemical Method

**DOI:** 10.3390/ma15227968

**Published:** 2022-11-11

**Authors:** Fabian N. Murrieta-Rico, Joel Antúnez-García, Rosario I. Yocupicio-Gaxiola, Jonathan Zamora, Armando Reyes-Serrato, Alexey Pestryakov, Vitalii Petranovskii

**Affiliations:** 1Ingeniería Mecatrónica, Universidad Politécnica de Baja California, Mexicali 21376, Mexico; 2Centro de Nanociencias y Nanotecnología, Universidad Nacional Autónoma de México, Ensenada 22800, Mexico; 3Departamento de Ingeniería Metalúrgica, Facultad de Química, Universidad Nacional Autónoma de México, Ciudad de México 04510, Mexico; 4Research School of Chemistry and Applied Biomedical Sciences, Tomsk Polytechnic University, Tomsk 634050, Russia; 5Laboratory of Catalytic and Biomedical Technologies, Sevastopol State University, Sevastopol 299053, Russia

**Keywords:** zeolite, MFI, iron, mechanochemistry, impedance spectroscopy

## Abstract

Zeolites are materials of undeniable importance for science and technology. Since the properties of zeolites can be tuned after the inclusion of additional chemical species into the zeolitic framework, it is necessary to study the nature of zeolites after modification with transition metals to understand the new properties that were obtained, and with this information, novel applications can be proposed. This paper reports a solvent-free approach for the rapid synthesis of zeolites modified with iron and/or iron oxide particles. The samples were characterized, and their electrical and magnetic properties were investigated.

## 1. Introduction

Zeolites are materials that have plenty of applications in diverse areas of science and technology [1]. Among other things, zeolites are used as adsorbents [2,3], ion exchangers [4], and catalyst supports [5,6]. Moreover, they can serve as sensitive materials for sensors [7]. By composition, zeolites are porous crystalline aluminosilicates that have interconnected channels. To date, there are more than 240 topologically different zeolite structures [8], each of which has a specific distribution of nanometric channels and cavities. The characteristic properties of zeolites make it easy to modify their chemical composition. Ion-exchange cations, located in the voids of their structure to neutralize the excess negative charge of the framework created by Al atoms, are easily exchanged for others. The isomorphic substitution of tetrahedrally coordinated atoms of Al and/or Si in the crystal structure also occurs rather easily [9,10]. Among various zeolite modifiers, iron attracts special attention [11]. Due to the inherent properties of iron and its variable valency, it can play the role of both an ion-exchange cation and/or replace atoms of the framework. In addition, iron easily forms oxide nanoparticles of various compositions on the surface of zeolites [12]. Among all known zeolitic structures, the MFI framework is used as an adsorbent or in the development of catalysts [13], which grants MFI zeolite an important place for research and industrial applications. Accordingly, it is desirable to understand the effects associated with the modification of MFI zeolite with metallic species such as iron.

Fe-containing zeolites are widely used in many catalytic applications, both because of the zeolite properties and due to the redox properties provided by the introduction of Fe compounds. It is noteworthy that after introducing Fe into the zeolite framework, the poor properties of conventional aluminosilicate zeolites, such as lack of active sites, low catalytic activity, poor anti-coking properties, etc., can be effectively improved [11]. The state of Fe in the zeolite, which significantly affects the properties of the obtained materials (stability, acidity, redox properties, etc.) of the Fe-zeolite, significantly depends on the method of introduction. The review [11] briefly outlines the latest developments in the field of synthesis, properties, and applications of Fe-based zeolites. Different species of iron in zeolites are attractive due to the variety of structures and functions, the relative ease of synthesis for a number of methods, economic efficiency due to the wide distribution of Fe in nature and its low cost, environmental friendliness, biocompatibility, and synergistic interaction with many other materials. The magnetic properties of some iron oxide nanoparticles make them much easier to clean and handle. These features make them promising catalysts for various chemical reactions [14].

Iron forms compound in several oxidation states. Some of the iron oxides and hydroxides, especially in the nanometer size range, are of great interest as heterogeneous catalysts, in particular in environmental remediation procedures using an advanced oxidation process that allows the efficient formation of reactive species such as hydroxyl radicals even at room temperature and at atmospheric pressure [13,15].

Catalysts play a crucial role in SCR technology, making it necessary to modify existing catalysts or develop new materials. A wide range of active temperatures from 150 to 650 °C is required, as well as exceptional N_2_ selectivity. The selective catalytic reduction of NO_x_ with NH_3_ over zeolite catalysts is recognized as the most practical techniques used in mobile vehicle exhaust to convert NO_x_ to N_2_ [16]. It should be noted that Fe-zeolite is being considered for commercial applications due to its advantages in fast SCR reactions. In addition, Fe-zeolite has been shown to provide higher NO_x_ conversion under real transient conditions [16].

Heinrich et al. [17] proposed a single-stage method for the preparation of Fe-containing zeolite with maximum ecological purity since the mechanochemical reaction was carried out in the absence of solvents. In contrast to the mentioned work, in the present study, we used not the protonated zeolite H-ZSM5 but its ammonium form NH_4_-ZSM5. The aim of our work was to confirm the possibility of the mechanochemical incorporation of iron into the composition of a zeolite sample and to characterize the material obtained by a number of physicochemical methods.

## 2. Materials and Methods

Zeolite MFI with a nominal SiO_2_/Al_2_O_3_ mole ratio of 50 in ammonium cation form (product CBV 5524G) and FeCl_3_ were purchased from Zeolyst International (KS, USA) and Sigma-Aldrich (MO, USA), respectively. According to the method of Heinrich et al. [17], the MFI zeolite was intensely ground in a mortar together with FeCl_3_ in a weight ratio of 2:1. The mixture was then placed in a beaker and stirred in water for 30 min; in this case, the excess of FeCl_3_ was dissolved. The solid precipitate was filtered off and washed with 500 mL of deionized water. The crystals were dried in an oven at 100 °C for 1 h. The chemical composition of the samples was evaluated by inductively coupled plasma-optical emission spectroscopy (ICP-OES) using a Vista-MPX CCD (CO, USA) simultaneous ICP-OES (Varian) (CO, USA). The samples were characterized by X-ray diffraction (XRD) using Aeris Panalytical (Malvern, UK) equipment with Cu K alpha monochromatic radiation (λ = 0.154056 nm, 40 kV, 15 mA) and UV-Vis spectroscopy using a UV-Vis NIR Cary 5000 (CA, USA) spectrophotometer. Micrographs of the samples were obtained using scanning electron microscopy (SEM) on JEOL JIB-4500 (MA, USA) equipment.

The textural properties of materials were determined by means of the nitrogen adsorption-desorption isotherms at −196 °C with ASAP 2000 equipment from Micromeritics (GA, USA). The samples were previously degassed at 300 °C for 4 h under vacuum before nitrogen adsorption, and two samples were pretreated in N_2_ flow at 160 °C and 300 °C. Surface area was measured with the Brunauer–Emmett–Teller (BET) method at relative pressures of 0.05 < P/P0 < 0.30. Average pore diameter was calculated following the Barrett–Joyner–Halenda method (BJH) using the desorption branch of the N_2_ isotherm. The cumulative pore volume ($V_{P}$) was obtained according to the amount adsorbed at a relative pressure of P/P0 = 0.99. The micropore surface area ($S_{micro}$) and the micropore volume ($V_{micro}$) were obtained with the t-plot method.

The XPS spectra of the samples were recorded using a SPECS^®^ spectrometer (Berlin, Germany) with a PHOIBOS^®^ 150 WAL hemispherical energy analyzer with angular resolution (<0.5°), equipped with an XR 50 X-Ray Al/Mg-X-ray anode and μ-FOCUS 500 X-ray monochromator (Al excitation line) sources.

In order to obtain electrochemical impedance spectroscopy (EIS) data, each sample was ground and compressed into a pellet with a diameter of 1 cm and thickness of 1 mm; the pellet was then placed between two polished copper electrodes, which were connected to E4980A Precision LCR Meter. As a result, for each sample, a data set was obtained corresponding to the magnitude of the total impedance $\left|Z_{T}\right|$ in ohms, phase angle θ in degrees, and frequency of interrogation signal in hertz. Magnetic measurements were performed using a Vibrating Sample Magnetometer (VSM) Squid Magnetometer Quantum Design MPMS^®^ 3 system (LA, USA) with a maximum applied field of 2.0 T at room temperature.

## 3. Results and Discussion

### 3.1. X-ray Diffraction

The experimental diffraction patterns obtained for the MFI and MFI-Fe samples are shown in Figure 1. For comparison, an XRD pattern obtained from the IZA database [8] corresponding to the standard MFI zeolite is shown. As can be observed, the experimental diffraction patterns of the original and iron-modified samples reveal the presence of characteristics peaks, corresponding to the reference data: although this is expected for a commercial zeolite (MFI), after modification with iron, this indicates the preservation of the topology of the zeolite structure. However, after the inclusion of iron in the composition of the sample, a slight shift to higher angles at the peak position can be observed. For example, the c lattice parameter undergoes a contraction from 20.15 Å to 20.05 Å with the inclusion of iron. This is consistent with the idea of a slight replacement of Al for Fe in the zeolitic structure, possible as an isomorphous substitution, due to the fact that Fe-O and Si-O bond lengths both are shorter than Al-O [18,19,20]. Therefore, its inclusion contributes to the contraction of the unit cell. On the other hand, we observe that the original zeolitic structure is retained after the mechanochemical modification with Fe. This fact is important because it is known that for some not-very-stable zeolites, such as LTA, as well as for too long mechanochemical treatment of other, stronger frameworks, the crystal structure of zeolites can be amorphized and destroyed.

For both samples, the degree of crystallinity or peak-to-noise ratio from XRD is presented in Table 1. In particular, the crystallinity is calculated from XRD by multiplying by 100 the division of area corresponding to crystalline peaks by the area of all peaks. In addition, from the Scherrer Equation (1), the average crystallites size $\bar{D}$ is calculated.
(1)$$D=\frac{K\lambda }{\beta \cos\theta }$$
where D is the crystallite size in nm, K=0.9 is the Scherrer constant, β is the line broadening at half the maximum intensity (FWHM), and λ=0.15406 is the wavelength in nm of the X-ray source.

### 3.2. Scanning Electron Microscopy and Energy Dispersive Spectroscopy

The morphology of MFI and MFI-Fe samples is presented in Figure 2. According to SEM micrographs, it can be noted that at a resolution of 5 μm, both samples, zeolite prior (Figure 2a) and after modification with iron chloride (Figure 2c), presents a quite similar grain morphology, but there are zones (see yellow circles in Figure 2a) that present a higher agglomeration on the MFI zeolite. Although in Figure 2c grain agglomeration is observed, agglomerated zones in Figure 2a could be attributed to large grains with small particles attached to them, while in Figure 2c, most of the grains are clearly defined. After a higher magnification, a similar level of homogeneity is observed for both samples (Figure 2b,c). In general, it can be noted from SEM micrographs that the average particle size slightly decreased as a result of mechanochemical modification using FeCl_3_.

The chemical composition of both samples was evaluated from EDS data. The normalized spectra of both samples are shown in Figure 3, which demonstrate the presence of elements typical for zeolites, such as Si, O, and Al. However, there are also substantive differences. Although Na is not present in the original sample, trace amounts of Na appear in the MFI-Fe sample after the mechanochemical process. It is assumed that this is a consequence of the presence of Na in the mortar material, and as a result of the grinding process, sodium could be integrated into the resulting material. Since EDS is a surface method of analysis, the appearance of a larger relative amount of Al on the surface, where characteristic X-rays are generated, could be due to the appearance of surface aluminum in an environment other than tetrahedral, namely, surface Al in an octahedral arrangement due to dealumination at the moment of crystal destruction. Although for MFI-Fe, the Fe peaks should appear at 0.705 and 6.398 eV for the Lα and Kα lines, respectively, its apparent absence could be attributed to low iron content. In general, terms, taking into account that the EDS data are generated by the characteristic X-ray radiation, which is emitted after atoms under study are impacted by electrons at a given penetration depth, from Figure 3, we can conclude that, after mechanochemical modification, the oxygen atoms of the MFI zeolite framework are rearranged and, as a consequence, the Al atoms are more exposed. This is caused by the level of particle agglomeration, as well as a decrease in the average crystallite size. In addition, since the O emission is increased, it could be concluded that oxides are present in greater amounts in the MFI-Fe sample. In other words, after the mechanochemical process, amorphization of a part of Al occurs, which is deposited on the surface. This raises the level of Al and O signals.

### 3.3. Inductively Coupled Plasma—Optical Emission Spectrometry

The results of the ICP-MS analysis are reported in Table 2. The SiO_2_/Al_2_O_3_ molar ratio was slightly above that specified by the supplier (SiO_2_/Al_2_O_3_). In passing, we note that the commercial sample of MFI contains an impurity of iron.

We cannot enter into a discussion of the reasons for such contamination; however, very often, the reagents used for synthesis are contaminated with traces of iron, which can change the composition of the final material. After our mechanochemical treatment by grinding in a mortar with ferric chloride, an increase in the concentration of iron by order of magnitude is observed. Moreover, a slight decrease in the amount of silicon in the material after the iron introduction was evident.

### 3.4. BET Analysis

After BET analysis, N_2_ adsorption isotherms were obtained (Figure 4). As is observed similar behavior is observed for both samples, which indicates that the main textural properties of the material are retained after modification. However, there is a difference in the absorbed quantity, which can be attributed to the difference in surface area as well as pore volume.

As can be seen from Table 3, the surface area of the MFI sample is larger than that of MFI-Fe, which indicates that there is a decrement in surface area after modification and, as a result, a lower adsorption capacity is observed. Thus, these measurements confirm the data of XRD analysis that the porous structure of the MPI zeolite is generally preserved during mechanochemical activation, and the observed decrease in the available pore volume is associated with the formation of nanoparticles of iron-containing compounds.

### 3.5. XPS Analysis

Finally, the XPS spectra for the MFI and MFI-Fe samples were obtained and analyzed. Previously, when analyzing the XRD data, it was noted that the experimental diffraction patterns indicate the preservation of the topology of the zeolite structure. However, a slight shift in the position of the peaks towards larger angles gave grounds to assume that there was a partial incorporation of Fe into the crystalline zeolite structure, which led to the compression of the unit cell. Analysis of the SEM data (Figure 2) showed that both samples have very similar grain morphology. The conducted EDS analysis did not reveal Fe peaks, which should appear at 0.705 and 6.398 eV for the Lα and Kα lines, respectively. Since the elemental composition data obtained by the ICP-OES method showed the presence of iron, the EDS results were explained by the fact that this is a surface analysis technique, in contrast to the ICP-OES method. The N_2_ adsorption-desorption isotherms for the MFI and MFI-Fe zeolites confirm the data of X-ray diffraction analysis that the porous structure of the MFI zeolite is generally preserved upon mechanochemical activation. The decrease in the available pore volume observed, in this case, is due to the fact that during mechanochemical treatment, nanoparticles of iron-containing compounds are formed in the voids of zeolite crystals due to ions diffusing into the channels of the zeolite structure.

Figure 5 shows high-resolution XPS spectra taken for elements such as Al, Si, and Fe. These data fully confirm the sum of observations obtained by other methods. The positions of the Si and Al peaks (103.01 and 74.47 eV, respectively) do not change since the structure of the MFI zeolite is preserved. There is no Fe peak in the spectra, which once again confirms the main result of this work. During mechanochemical activation, there is no “spreading” of iron compounds over the surface of zeolite crystals. Iron ions, on the contrary, penetrate into their volume, where some of them are introduced into the crystal structure, as evidenced by XRD data, and some are hydrolyzed with the formation of oxide-hydroxide nanoparticles in the channels, which is confirmed by adsorption data. These iron compounds are responsible for the magnetic properties of the samples.

### 3.6. Band-Gap Analysis

The Band gap of zeolitic nanoparticles was obtained using the UV–Vis spectra of the samples under study in absorbance mode and following the Tauc model:(2)$${\left(ahv\right)}^{1/n}=A\left(hv-E_{g}\right)$$
where h is Planck’s constant, *v* is the frequency of the photon, *α* is the adsorption coefficient, $E_{g}$ is the band gap, and A is the slope of the Tauc graph in the linear region. The data of linear fitting are presented in Figure 6. The band gap of the MFI zeolite is modified after the inclusion of Fe. In fact, the band gap of the MFI-Fe sample lies between the band gaps of MFI and FeCl_3_.

### 3.7. Electrochemical Impedance Spectroscopy

It can be seen from the EIS data that MFI and MFI-Fe have almost the same frequency response (Figure 7a,b); at low frequencies (Figure 7c), the DC conductivity $\sigma_{dc}$ is higher for the MFI-Fe sample; in contrast, at high-frequency values, the AC conductivity $\sigma_{ac}$ is greater for the MFI sample. These plots show how the conductivity of the MFI zeolite can be tuned after the addition of Fe, which is observed in the Argand diagram (Figure 7d). According to Jonscher’s power law for, the universal behavior of conductivity is given by
(3)$$\sigma_{ac}=\sigma_{dc}+A_{j}\omega^{n},$$
where $A_{j}$ is the pre-exponential factor, ω is the angular frequency in rad/s, and n is the frequency exponent. Using the data in Figure 7c, the parameters of Equation (3) ($\sigma_{dc}$, $A_{j}$, n) can be fitted, and the parameters of the elements in Joshcher’s power law are found for each sample.

The behavior of the total electrical conductivity $\sigma_{T}$ is quite similar, this is illustrated by the parameters of Joshcher’s power law (Table 4). In particular, the value of *n* for both samples is equal to 1, which is typical for ionic compounds. In addition, the value of n defines the transition from DC to AC conductivity, which in this case, is observed to be the same. The differences in $\sigma_{T}$ can be attributed to $\sigma_{ac}$ and how the frequency ω is escalated by $A_{j}$ for each sample. The Argand diagram shows the relationship between the real and imaginary parts of the impedance. Given that
(4)$$Z_{T}=Z^{\prime }+jZ^{\prime }$$
and
(5)$$Z^{\prime }=\left|Z_{T}\right|\cos\theta ,$$
(6)$$Z^{\prime \prime }=\left|Z_{T}\right|\sin\theta ,$$

It can be noted for both samples that at low frequencies (Figure 7d), the value of $Z^{\prime }$ is greater, there is a positive slope, and after a local minimum, there is an increment describing a semicircle. In the case of MFI, there is an absolute maximum, and in the case of MFI-Fe, there is a local maximum. Both samples have a decrement with a final value at the origin. Since Equation (4) apparently defines total impedance in terms of a resistive $Z^{\prime }$ and reactive $Z^{\prime \prime }$ parts, it could be considered that in the Argand diagram, the reactive effects have a greater influence in the MFI sample at higher frequencies. Alike in MFI zeolite, after modification with Fe, MFI-Fe shows more reactive effects at low frequencies. These observations can be attributed to a similar ionic conductivity mechanism but with a different contribution of each element involved in such a process.

### 3.8. Magnetic Characterization

Figure 8 shows the hysteresis loops at 300 K for MFI and MFI-Fe samples. For the MFI sample in Figure 8a, an anti-S-type hysteresis curve is observed, which indicates a principal diamagnetic behavior caused by the matrix and the sample holder. The inset shows the ferromagnetic-like behavior after subtracting the diamagnetic component. Meanwhile, the Fe-modified MFI sample, see Figure 8b, exhibits an S-type hysteresis curve, i.e., presents weak ferromagnetic (FM) behavior with a coercivity around 600 Oe and very small magnetization values without reaching saturation, according to the response and magnetic behavior for this type of materials [21]. This FM behavior can be associated with the presence of oxy-hydroxides; however, it can also be related to an existing ferrimagnetic order. However, both samples exhibit small magnetization values, which may be due to the influence of preferred orientation in a single axis or the presence of a small phase fraction causing a magnetic dilution effect leading to the rather low magnetization, which is also consistent with the ICP analysis.

## 4. Conclusions

This article reports a solvent-free approach to the rapid one-step synthesis of zeolites modified with iron and/or iron oxide particles. The modification of the MFI zeolite with iron was carried out by the mechanochemical method. As a result, it was observed that the zeolite, after modification, retains its original crystalline structure; however, there is a variation in physicochemical properties, such as the band gap. Moreover, as a result of the introduction of Fe into the composition of the sample, a weak ferromagnetic behavior was observed. This effect causes a change in the electrical properties of the zeolite, which is observed as a decrease in zeolite reactance.

## Figures and Tables

**Figure 1 materials-15-07968-f001:**
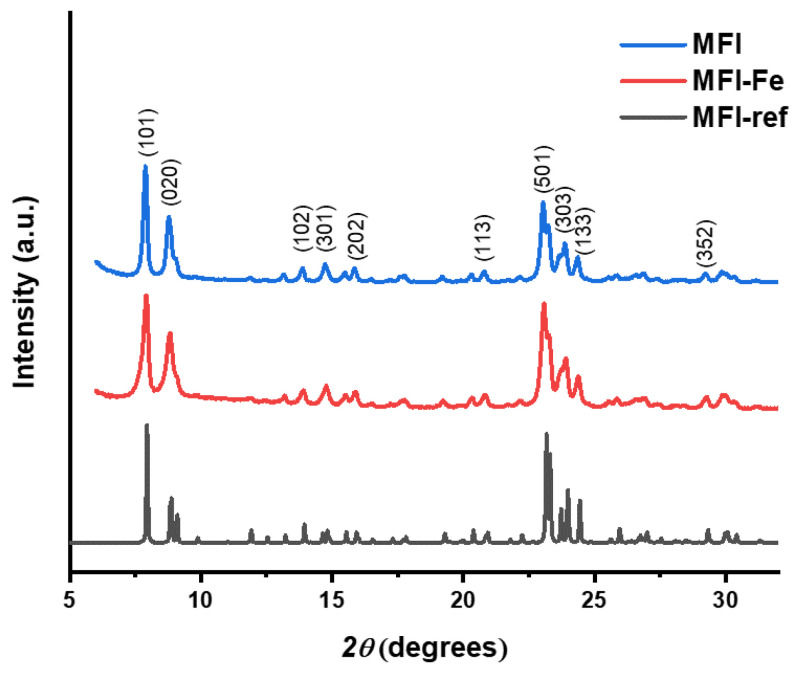
Diffractograms of MFI zeolite, iron-modified zeolite (MFI-Fe), and reference ZSM-5 sample from the IZA database [8] (MFI-ref).

**Figure 2 materials-15-07968-f002:**
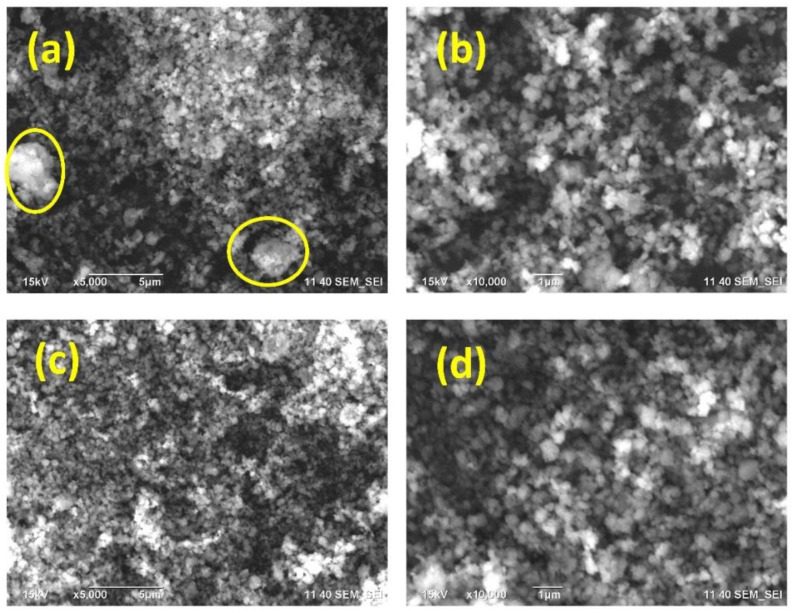
Micrographs of zeolites: MFI with a resolution of 5 μm (**a**) and 1 μm (**b**); and MFI-Fe with a resolution of 5 μm (**c**) and 1 μm (**d**).

**Figure 3 materials-15-07968-f003:**
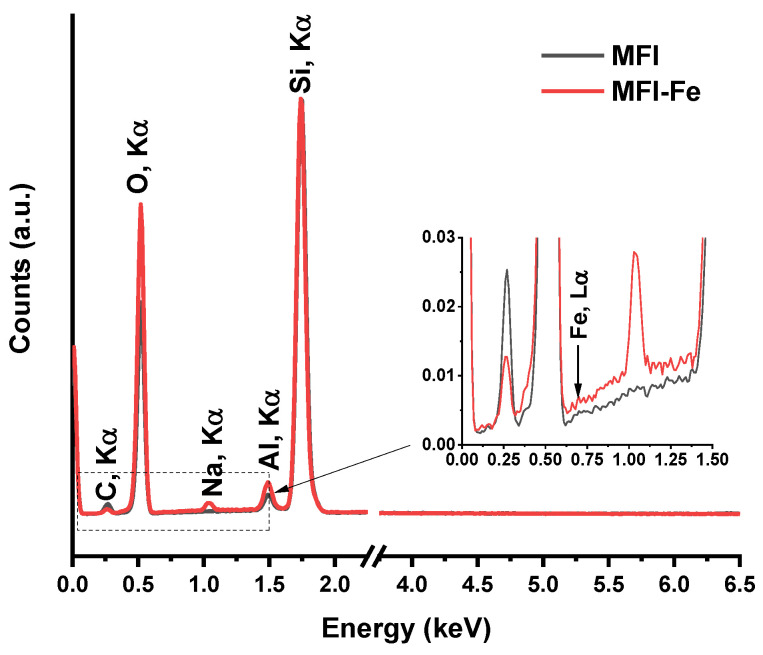
Data of EDS (Energy-dispersive X-ray spectroscopy) for both samples.

**Figure 4 materials-15-07968-f004:**
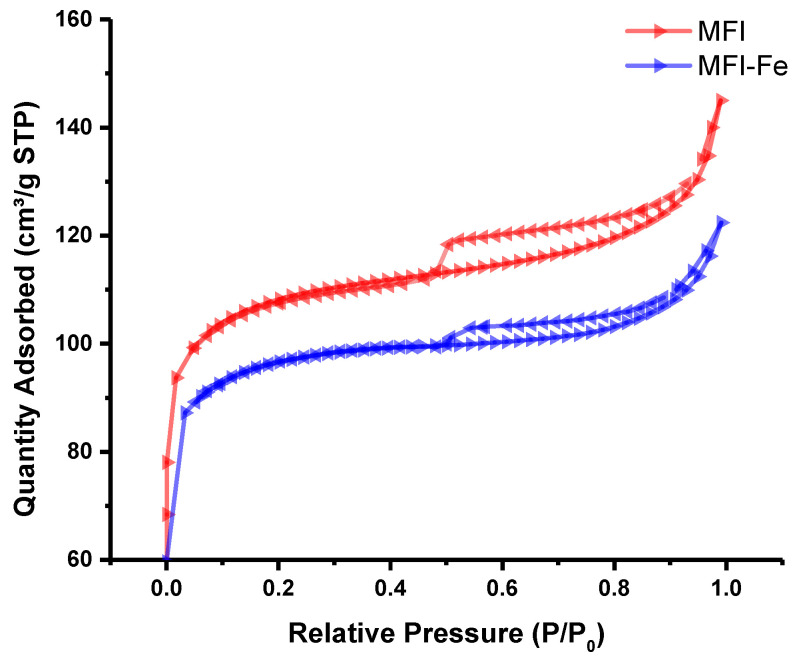
N_2_ adsorption-desorption isotherms for MFI and MFI-Fe zeolites.

**Figure 5 materials-15-07968-f005:**
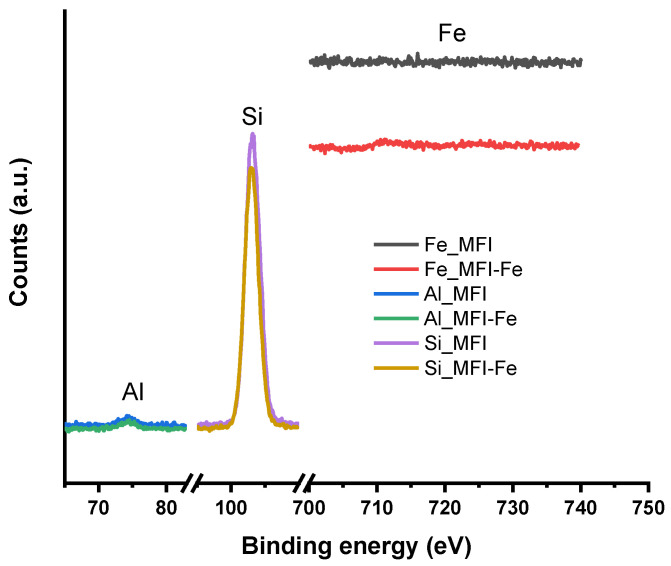
XPS spectra in the BE ranges of Al, Si, and Fe for MFI and MFI-Fe zeolites.

**Figure 6 materials-15-07968-f006:**
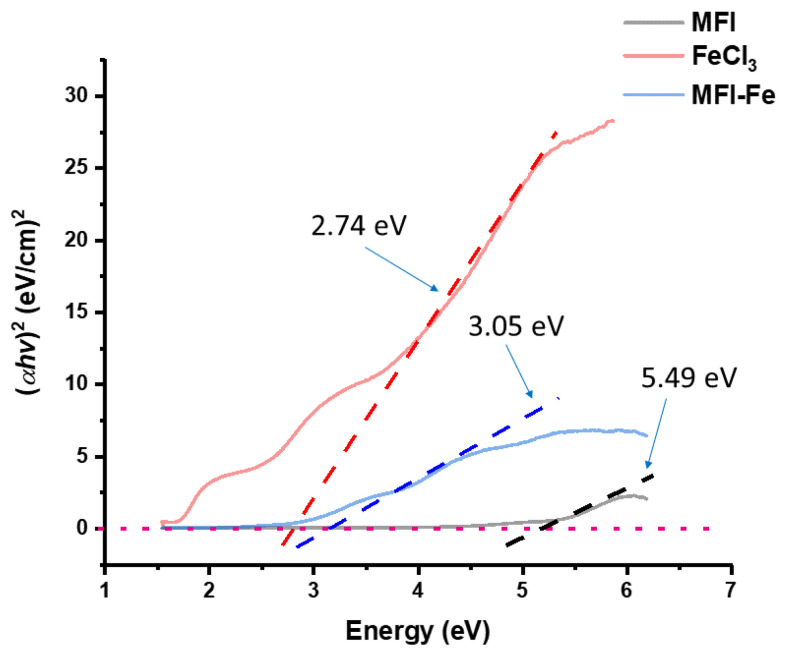
Tauc plot for band-gap characterization of pure MFI zeolite, iron-chloride, and iron-modified MFI zeolite.

**Figure 7 materials-15-07968-f007:**
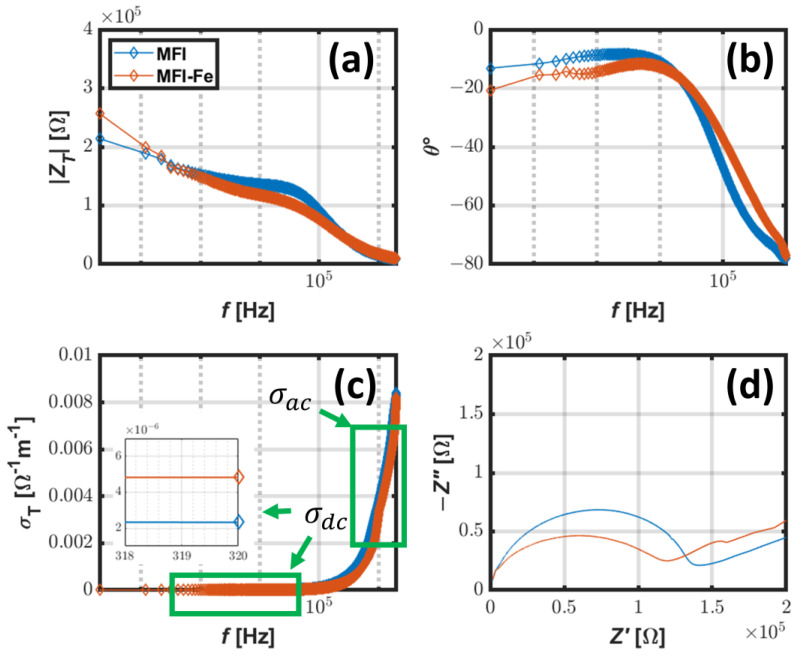
The magnitude of total impedance $\left|Z_{T}\right|$ (**a**); phase angle θ (**b**); total conductivity $\sigma_{T}$ (**c**), and Argand diagram of the real part of impedance $Z^{\prime }$ vs. imaginary part of impedance Z″ (**d**).

**Figure 8 materials-15-07968-f008:**
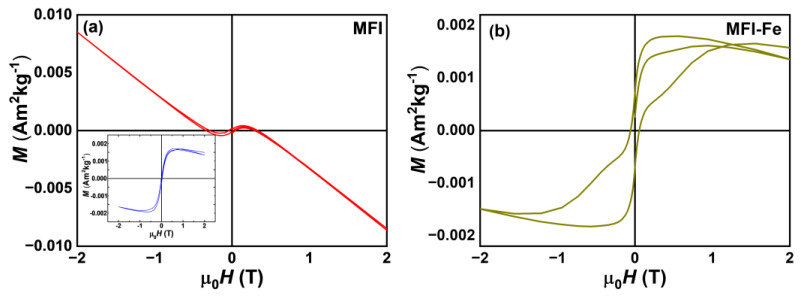
Hysteresis loops of both zeolites at room temperature; (**a**) MFI, inset: corrected for diamagnetic background, and (**b**) Fe-modified MFI sample.

**Table 1 materials-15-07968-t001:** Analysis of average crystallite size and crystallinity of MFI and MFI-Fe samples.

Sample	Crystallinity	$\bar{D}$ (nm)
MFI	93.55%	19
MFI-Fe	98.12%	15

**Table 2 materials-15-07968-t002:** Elemental composition data from ICP-OES for MFI and MFI-Fe zeolites.

Sample	Al_2_O_3_ % Mole	SiO_2_ % Mole	Fe_2_O_3_ % Mole	NH_4_ % Mole	SiO_2_/Al_2_O_3_ Mole Ratio	SiO_2_/Fe_2_O_3_ Mole Ratio	SiO_2_/(Al_2_O_3_ + Fe_2_O_3_) Mole Ratio
MFI	1.52	95.34	0.09	3.05	62.54	1045.49	59.01
MFI-Fe	1.55	94.67	0.70	3.09	61.25	136.02	42.23

**Table 3 materials-15-07968-t003:** Pore textural properties of MFI and MFI-Fe zeolites.

Sample	$V_{p}$ (cm^3^/g)	$S_{BET}$ (m^2^/g)
MFI	0.14	322
MFI-Fe	0.11	286

**Table 4 materials-15-07968-t004:** Values obtained from fitting Jonscher’s power law.

Sample	$\sigma_{dc}$	$A_{j}$	n
MFI	$1.5\times 10^{-6}$	$3.86\times 10^{-9}$	1
MFI-Fe	$2.5\times 10^{-6}$	$3.37\times 10^{-9}$	1

## Data Availability

Not applicable.

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
