# Peer review of "Study of Electric and Magnetic Properties of Iron-Modified MFI Zeolite Prepared by a Mechanochemical Method"

_materials, 2022, doi:10.3390/ma15227968_

Round 1

Reviewer 1 Report

In this work, the authors present the preparation of Fe-MFI zeolite via a mechanochemical method, and their electrical and magnetic properties were investigated. However, the overall contents in the manuscript are not fascinating. This synthetic method and Fe-based zeolite material are the same as the reported literature (Catalysis Communications 2(2001)317), and the relationship between their structure and the electrical and magnetic properties could not evidence the corresponding applications. Therefore, I cannot recommend this paper for publication in Membranes.

Author Response

Please, find the attached file

Reviewer 2 Report

In this manuscript, the author studied the magnetic and electrical properties of iron-modified MFI zeolites which were prepared by a mechanochemical method. The manuscript needs several improvements. However, there are some comments which may be helpful to improve the manuscript.

[1]   The MFI zeolite is modified through the mechanochemical method; thus, structural change is expected. The author mentioned in XRD data: ‘quite similar diffractogram is observed after modification with iron’. How it is possible? Justify it. If MIF is modified then there should be a slight change in the XRD pattern (with some change in angle). Is there no change in 2θ angle? If yes, then mention the angle before and after the modification. If not, then how the author can justify the modification? Also, mention the planes and their arrangement based on XRD data.

[2]   Zeolite absorbs the moisture from the air into its pores. The water molecules get trapped inside the pores. Moisture is expelled from the zeolite after heating it. All the SEM images are not clear, which may be due to the presence of moisture in the samples. The large agglomeration may also be due to the moisture content. The author should again analyze the samples. Revise the justification carefully.

Author Response

Please, find the attached file

Reviewer 3 Report

In this manuscript, the authors explored the electric and magnetic properties of iron-modified MFI 2 zeolite prepared by a mechanochemical method. This manuscript may be of interest to the readers of the field, working in this area. Therefore, I recommend this manuscript for publication but some changes may be made to further improve the quality of the manuscript. 

1. An Arrhenius plot may be included to calculate the activation energy and discuss the conduction mechanisms across the samples.

2. The results obtained from the Arrhenius plot may also be compared with the UV-data

3. A table for the elemental analysis using EDS may be included.

Author Response

Please, find the attached file

Round 2

Reviewer 1 Report

Authors have not addressed my suggestions/comments, and should further explain in the revised manuscript the significance and novelty of the work and the problem that is being addressed. The overall quality of the manuscript has not considerably improved. Therefore, this current manuscript is not appropriate for publication by Materials.

Author Response

Please, find an attached file

Reviewer 2 Report

May be accepted in the present form as authors have modified the manuscript as per suggestions.

Author Response

We thank the reviewer for a good assessment of our work.